# Characterization of PD-1/PD-L1 Immune Checkpoint Expression in Osteosarcoma

**DOI:** 10.3390/diagnostics10080528

**Published:** 2020-07-29

**Authors:** Kazuhiko Hashimoto, Shunji Nishimura, Masao Akagi

**Affiliations:** Department of Orthopedic Surgery, Kindai University Hospital, 377-2 Ohno-Higashi, Osaka-Sayama City, Osaka 589-8511, Japan; shunnisi@med.kindai.ac.jp (S.N.); makagi@med.kindai.ac.jp (M.A.)

**Keywords:** osteosarcoma, PD-1, PD-L1, immune checkpoint pathway

## Abstract

Recent data have suggested that PD-1 and PD-L1 are involved in osteosarcoma (OS) pathogenesis; however, their contributions are not well-established. Here, the PD-1/PD-L1 expression was evaluated in (OS) cases. Preoperative needle biopsy specimens were obtained from 16 patients with OS. Immunostaining for CD4, CD8, PD-1, and PD-L1 was performed on pathological specimens. Clinical parameters, including age, tumor size, preoperative alkaline phosphatase (ALP) level, standardized uptake value (SUV)-max level, and survival rate, were compared between PD-1/PD-L1-positive and -negative groups. CD4-, CD8-, PD-1-, and PD-L1-positive rates among all specimens were 75%, 75%, 18.7%, and 62.5%, respectively. The rates of co-expression of CD4 and CD8 with PD-L1 were 56.2% and 50%, respectively. Tumors were significantly larger in PD-L1-negative cases than in PD-L1-positive cases. Age, size and ALP and SUV-max levels did not differ significantly between PD-1/PD-L1-positive and -negative cases. The 3-year survival rates did not differ significantly between PD-1-positive and -negative cases or between PD-L1-positive and -negative cases. However, the occurrence of cancer-related events, including recurrence, metastasis, and death was associated with the PD-1-negative and PD-L1-positive status. The PD-1/PD-L1 checkpoint is likely involved in the immune microenvironment in OS and may be involved in the occurrence of cancer-related events.

## 1. Introduction

Osteosarcomas (OSs) are derived from primitive mesenchymal cells. They are malignant bone tumors that primarily originate from the bone marrow, and rarely from soft tissue. [1]. Despite major advances in diagnosis and clinical treatment, the prognosis for patients with osteosarcoma remains poor, with a 5-year survival rate of around 30% [2]. Although 70% of patients with localized disease benefit from chemotherapy and surgical resection, patients with metastatic OS are typically refractory to treatment [3]. Therefore, novel and effective therapies for sarcomas are urgently needed for patients with metastasis or unresectable malignancies [4]. Recently, immune checkpoint inhibitors have been used to alleviate the immunosuppressive state of the tumor microenvironment in solid malignancies by restoring the immune function of T cells and killing tumor cells [5]. Evidence for the involvement of the PD-1/PD-L1 immune checkpoint pathway in OS is emerging [4,5,6]. However, evidence for the roles of PD-1/PD-L1 in OS is still insufficient. In the present study, we used osteosarcoma specimens from cases treated at our hospital to investigate and characterize the relationship between clinical factors and the expression status of PD-1/PD-L1 immune checkpoint proteins, including CD4 and CD8.

## 2. Materials and Methods

Patients with OS treated at our hospital from January 2009 to June 2018 were retrospectively reviewed. The current study was approved by the Kindai University Ethics Committee (approval number: 31-187; Approved on 16 January 2020). Cases in which the clinical content of the case or the final outcome could not be followed were excluded. Age, sex, tumor size, ALP level, SUV-max level, treatment method, recurrence, metastasis, clinical outcome, and the 3-year survival rate were surveyed. The tumor size was calculated as the length of the vertical axis of the maximum plane × the length of the horizontal axis.

Immunostaining was performed using specimens from needle biopsies of patients with OS treated at our institution. Tissues were deparaffinized, rehydrated, and subjected to antigen retrieval using 3% hydrogen peroxide. The sections were incubated with the following primary antibodies: CD4 antibody (rabbit monoclonal, Roche Diagnostics, Risch-Rotkreuz, Switzerland) for 32 min at 37 °C after 60 min high pH (9.0) heat activation; CD8 antibody (rabbit monoclonal, Nichirei Corporation, Tokyo, Japan) for 32 min at 37 °C after 60 min high pH (9.0) heat activation; PD-1 (mouse monoclonal, ab52587; Abcam, Cambridge, UK) for 30 min at 37 °C after 30 min low pH (6.0) heat activation; and PD-L1 antibody (rabbit monoclonal, ab205921; Abcam) for 32 min at 37 °C after 60 min high pH (9.0) heat activation. Sections were incubated with the corresponding secondary antibodies for 30 min at 37 °C. The reaction was visualized using 3,3′-diaminobenzidine (DAB) (DAB Substrate Chromogen System; DAKO, Kyoto, Japan) and counterstained with hematoxylin. Slides were observed under a microscope (BIOREVO BZ-9000; KEYENCE, Osaka, Japan) and brown granules in the cytoplasm or nuclei indicated positive staining. Positive staining rates for each immunostaining target (CD4, CD8, PD-1, and PD-L1) were calculated. The percentages of cases positive for CD4, CD8, and PD-1 matched with PD-L1-positive cases were examined. Differences in age, tumor size, preoperative ALP, and mean SUV-max levels between the PD-1/PD-L1-positive and -negative groups were evaluated. The 3-year survival rates in groups positive or negative for each factor were calculated. Finally, PD-1 and PD-L1 staining characteristics in cases with worse outcomes, including cases with recurrence, metastasis, and cancer-related death, were evaluated.

### Statistical Analyses

Student’s *t*-tests were used for comparisons between two groups. The Kaplan–Meier method was used to evaluate the 3-year survival rate, and log-rank tests were used for comparisons between groups. A value of *p* < 0.05 was considered significant. Analyses were implemented in Stat Mate 5.05 (ATMS, Tokyo, Japan).

## 3. Results

Patient characteristics are summarized in Table 1. The mean follow-up period was 39.5 months (range, 10–165 months). The mean age of patients was 28 years (range, 8–74 years). Nine males and seven females were included.

Ten cases of conventional OS, three cases of chondroblastic OS, two cases of fibroblastic OS, and one case of osteoblastic OS were included. One case was classified as stage IA, seven as stage IIA, six as stage IIB, one as stage III, and one as stage IVB. Chemotherapy and operation were performed in 13 cases, chemotherapy and heavy particle radiation were performed in two cases, and operation was performed in one case.

Recurrence and metastasis were each observed in four patients. With respect to clinical outcomes, nine patients were continuously disease-free, one showed no evidence of disease, two were alive with disease, and four were dead of disease (DOD).

The overall positive rates were 12/16 (75%) for CD4, 12/16 (75%) for CD8, 3/16 (18.7%) for PD-1, and 10/16 (62.5%) for PD-L1. The CD4-positive and PD-L1-positive rate was 9/16 (56.2%). The CD8-positive and PD-L1-positive rate was 8/16 (50%). The PD-1 and PD-L1 concordance rate was 3/16 (18.7%). Representative positive or negative histological findings are shown in Figure 1a–h.

We also confirmed the localization of CD4, CD8, and PD-1 stained cells using adjacent section specimens (Figure 2A–C). Tumor-infiltrated lymphocytes (TILs) were found to be positive for CD4, CD8, and PD-1 (Figure 2A–C).

Patient ages did not differ significantly between the PD-1-positive group (36 ± 33.1 years; *n* = 3, Table 2) and the PD-1-negative group (28 ± 16.4 years; *n* = 13, Table 2) (*p* = 0.53). Patient ages were 32 ± 22.3 years in the PD-L1-positive group (*n* = 10, Table 2) and 26 ± 13.9 years in the PD-L1-negative group (*n* = 6, Table 2), and the difference was not significant (*p* = 0.56).

Tumor sizes were 12 ± 15.5 cm in the PD-1-positive group (*n* = 3, Table 2) and 15.5 ± 30.0 cm in the PD-1-negative group (*n* = 10, Table 2), and the difference was not significant (*p* = 0.85). Tumor sizes were 26.7 ± 35.3 cm in the PD-L1-positive group (*n* = 10, Table 2) and 58.5 ± 20.2 cm in the PD-L1-negative group (*n* = 6, Table 2), and the difference was not significant (*p* = 0.065).

Alkaline phosphatase (ALP) levels were 273 ± 199.3 U/L in the PD-1-positive group (*n* = 3, Table 2) and 363.5 ± 294.5 U/L in the PD-1-negative group (*n* = 10, Table 2), and the difference was not significant (*p* = 0.63). ALP levels were 266 ± 278.9 U/L in the PD-L1-positive group (*n* = 8, Table 2) and 368 ± 274.7 U/L in the PD-L1-negative group (*n* = 5, Table 2), and the difference was not significant (*p* = 0.53).

SUV-max levels were 12.7 ± 6.56 in the PD-1-positive group (*n* = 3, Table 2) and 10.5 ± 5.47 in the PD-1-negative group (*n* = 5, Table 2), and the difference was not significant (*p* = 0.62). SUV-max levels were 11.1 ± 5.51 and 11.3 ± 1.15 in the PD-L1-positive (*n* = 8, Table 2) and negative groups (*n* = 2, Table 2), respectively. The slight difference observed was not statistically significant (*p* = 0.48).

The 3-year survival rates were 100% for PD-1-positive cases and 64.1% for PD-1-negative cases (Figure 3A), and the difference was not significant (*p* = 0.38). The 3-year survival rates were 60% for PD-L1-positive cases and 83.3% for PD-L1-negative (Figure 3B), and the difference was not significant (*p* = 0.49).

The 3-year survival rate for PD-L1-positive cases was 60% and for negative cases was 83.3%; the difference was not significant (*p* = 0.49).

In cases with recurrence, 3/4 (75%) were PD-1-negative and 3/4 (75%) were PD-L1-positive (Table 1). In cases with metastasis, 4/4 (100%) were PD-1-negative and 2/4 (50%) were PD-L1-positive. Among DOD cases, 4/4 (100%) were PD-1-negative and 3/4 (75%) were PD-L1-positive.

## 4. Discussion

Scattered pieces of evidence suggest the involvement of the PD-1/PD-L1 immune checkpoint in the pathogenesis of OS; however, the evidence is inconclusive. In the current study, we characterized the expression of PD-1/PD-L1 immune checkpoint proteins with respect to clinical features in patients with OS, using immunohistochemistry. This is the first detailed examination of PD-1/PD-L1 expression, including CD4 and CD8, from a clinical perspective, because clinical parameters have not previously been evaluated.

CD4-positive T cells and CD8-positive T cells are involved in the tumor immune environment in OS [7]. CD8-positive T cells have been reported to have a greater role in the tumor immune environment than CD4-positive T cells [7]. CD8+ T cells inhibit the development of metastasis in a mouse OS model by a mechanism that requires the production of IFN-γ (Interferon-γ) [8]. In the current study, we observed the same positive rate of CD4+ TILs with CD8 TILs. CD4-positive and CD8-positive TILs are thought to have similar contributions to tumor immunity in osteosarcoma.

A previous study reported that the PD-1-positive T cell rate is approximately 20%, similar to the rate in the current study [9]. The reported PD-1-positive rate in OS is 47% [10]. A high level of expression of PD-L1 has been observed in 23.7% of OS tumors, and an intermediate level of expression was reported in 50% of tumors [11]. The positive rate of PD-L1 in OS cases was previously reported to be 53% [10]. Although the percentage of positive cells in tumors was not calculated in this study, because the study was based on biopsy specimens, a high percentage of PD-L1-positive tumor cells were found. These findings suggest that the PD-1/PD-L1 system plays an important role in tumor immunity in OS. The numbers of CD8-positive TILs and PD-L1-positive tumor cells are correlated in the tumor immune environment of OS [11]. A previous study has shown that the percentage of PD-1-positive cells is significantly elevated in both peripheral CD4-positive and CD8-positive T cells in patients with OS [12]. In the current study, strong correlation between CD4 positivity or CD8 positivity and PD-L1 were observed. These findings also indicate that CD4-positive T cells, CD8-positive T cells, PD-1, and PD-L1 are involved in the tumor immune environment in OS.

Previous studies have shown that the proliferation ability of CD4-positive T cells decreases with age [13,14]. It has recently been reported that PD-1 expression in CD4-positive T cells increases with age in mice. These cell populations (CD4-positive cells with PD-1) lose their ability to respond to antigenic stimuli [15]. Studies using human sarcoma specimens have confirmed that PD-L1 expression is elevated in individuals over 65 years of age [16]. Although no differences in age were detected between PD-1-positive and -negative or PD-L1-positive and -negative cases, the only patient over 65 years of age in this study, a 74-year-old man, was positive for all markers: CD4, CD8, PD-1, and PD-L1. Age is a poor prognostic factor in OS [17], suggesting that PD-1/PD-L1 immune checkpoint activity was lost with age in the 74-year-old patient.

In general, tumor size is an important prognostic factor in OS [18]. Tumor sizes greater than 16 cm are associated with a significantly poorer 5-year survival rates than smaller tumors [19]. In the current study, no significant difference in tumor size was observed between PD-1-positive and -negative cases or between PD-L1-positive and -negative cases.

ALP levels indicate the extent of bone destruction in OS, and ALP has been reported to be negatively associated with survival [20]. A recent meta-analysis has reported that ALP is not a prognostic factor, suggesting that its role is controversial [21]. In the present study, there were no significant differences in ALP levels between patients with and without PD-1/PD-L1 expression.

In a study using FDG-PET, the SUV-max value was identified as a prognostic factor in OS [22]. FDG-PET imaging of OS positively correlates with the histologic response to neoadjuvant chemotherapy [23]. In the current study, there was no significant difference in SUV-max values between PD-1-positive and -negative cases or between PD-L1-positive and -negative cases.

In a recent report, PD-L1 expression correlated with immune cell infiltration and event-free survival [24]. The authors also suggested that PD-L1 expression is significantly associated with a poorer 5-year event-free survival [24]. PD-1 and PD-L1 expression are associated with advanced and metastatic disease, high grade disease, differentiation, tumor necrosis, and shorter disease-free and overall survival [25].

It has also been reported that a high level of PD-L1 predicts a reduced 5-year event-free survival in patients with OS, and is correlated with early metastasis [24,26]. We did not observe a significant difference in survival rates between PD-1-positive and -negative cases or PD-L1-positive and -negative cases. However, 75% of cases of local recurrence were PD-1-negative and PD-L1-positive. In addition, 100% of metastatic cases were PD-1-negative, and 50% were PD-L1-positive. Moreover, 100% of patients who died were negative for PD-1 and 75% were positive for PD-L1. These findings indicate that, in OS, a PD-1-negative and PD-L1-positive status may be associated with the occurrence of cancer-related events, such as recurrence, metastasis, and death.

In conclusion, we surveyed and characterized the expression of CD4, CD8, PD-1, and PD-L1 in OS cases treated in our hospital in detail. We believe that the PD-1/PD-L1 immune checkpoint system, involving CD4 and CD8, plays an important role in the pathogenesis of OS and a PD-1-negative with PD-L1-positive status may predict an event occurrence.

## Figures and Tables

**Figure 1 diagnostics-10-00528-f001:**
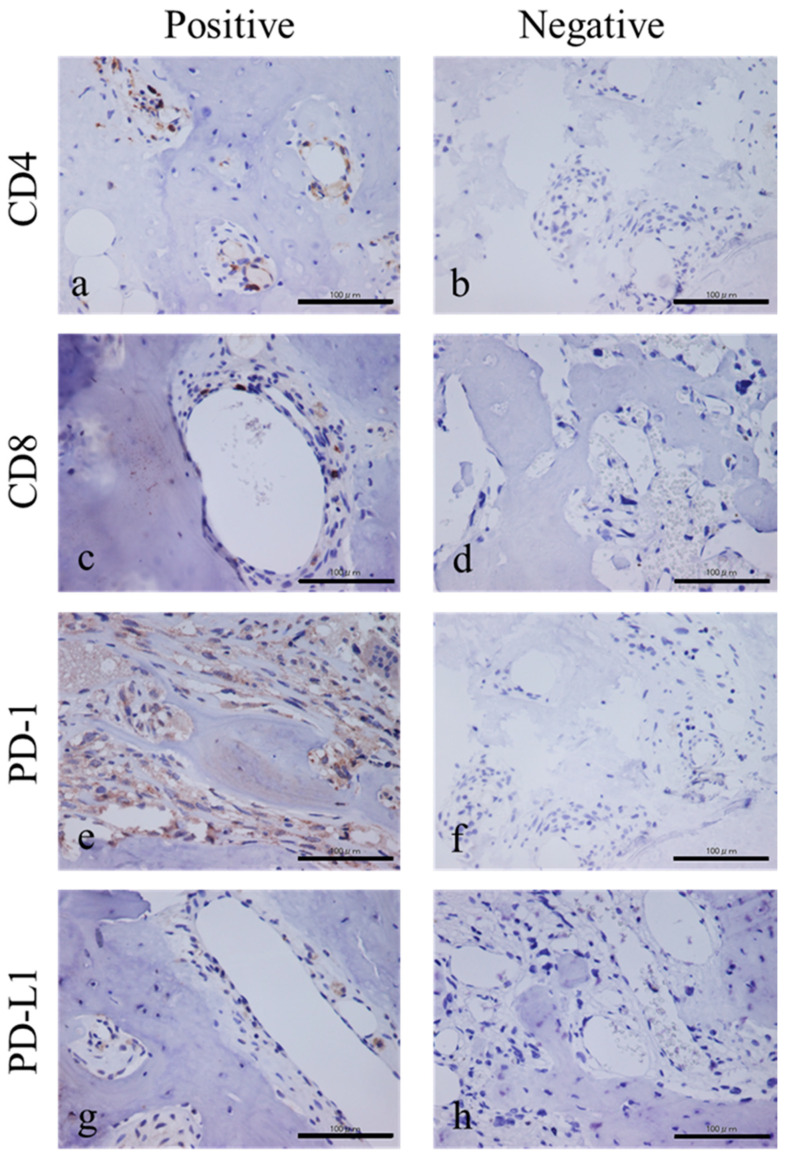
Representative histopathological findings (**a**–**h**). CD4-positive histological findings (**a**) and CD4-negative findings (**b**). CD8-positive histological findings (**c**) and CD8-negative findings (**d**). PD-1-positive histological findings (**e**) and PD-1-negative findings (**f**). PD-L1-positive histological findings (**g**) and PD-L1 negative findings (**h**).

**Figure 2 diagnostics-10-00528-f002:**
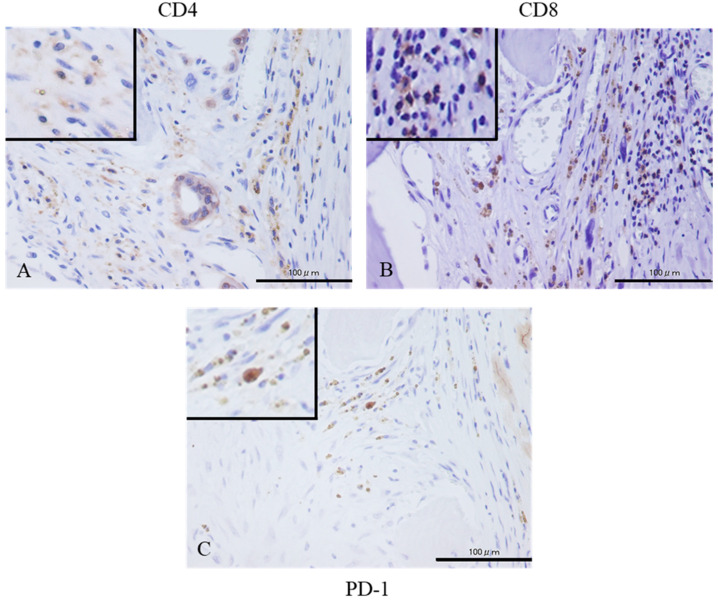
The larger microscopic view of CD4 (**A**), CD8 (**B**), and PD-1 (**C**) immunostaining (magnification. Immunostaining results of CD4, CD8, and PD-1 in adjacent sections are shown 400×) Tumor-infiltrated lymphocytes (TILs) present at the same or nearby site can be seen in each stain. The inset in the upper left corner of each image is an enlarged image, showing CD4, CD8, PD-1-positive TILs. Scales bar = 100 µm.

**Figure 3 diagnostics-10-00528-f003:**
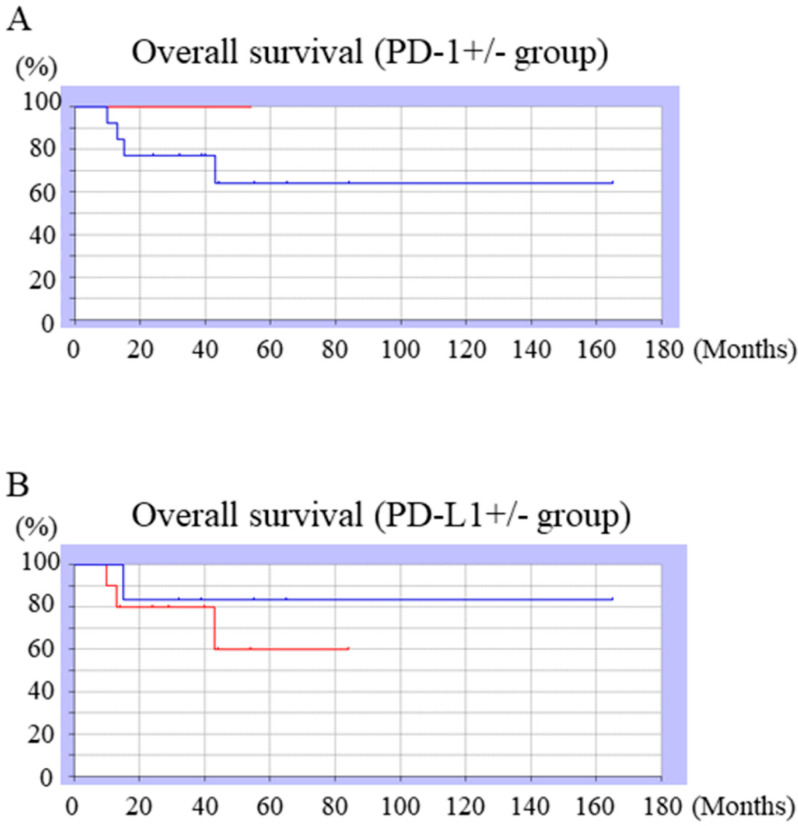
(**A**) The 3-year survival rate for PD-1-positive (blue line) and -negative cases (red line). The 3-year survival rate for PD-1 positive cases was 100% and that for negative cases was 64.1%, and the difference was not significant (*p* = 0.38); (**B**) The 3-year survival rate for PD-L1-positive (red line) and -negative cases (blue line).

**Table 1 diagnostics-10-00528-t001:** Clinical features of patients with primary osteosarcoma treated at our hospital and immunopositivity for CD4, CD8, PD-1, and PD-L1.

Patient No.	Age (y)/Sex	Site	Stage	Treatment	Chemotherapy	Local Recurrence	Metastasis	Follow-Up (mo)	Outcome	CD4	CD8	PD-1	PD-L1
1	17/M	Tibia	IVB	CT, WR	NECO-95J	+	+	43	DOD	+	+	-	+
2	8/M	Humerus	III	CT, WR	NECO-95J	-	+	29	CDF	+	+	+	+
3	52/M	Pelvis	IIB	CT, WR	NECO-95J	+	-	32	AWD	+	+	-	-
4	14/M	Tibia	IIA	CT, WR	NECO-95J	-	-	24	CDF	+	-	-	+
5	13/F	Femur	IIB	CT, WR	NECO-95J	-	-	65	CDF	+	+	-	-
6	15/F	Femur	IIA	CT, WR	NECO-95J	-	-	84	CDF	+	+	-	+
7	36/F	Tibia	IIA	CT, WR	NECO-95J	-	-	40	CDF	+	+	-	+
8	51/M	Humerus	IA	CT, MR	IA	-	-	44	CDF	+	+	-	+
9	74/M	Radius	IIB	CT, WR	IA × 80%	-	-	14	CDF	+	+	+	+
10	16/M	Tibia	IIB	CT, WR	NECO-95J	-	-	39	CDF	-	-	-	-
11	36/F	Tibia	IIA	CT, MR	NECO-95J	+	-	54	NED	+	-	+	+
12	24/M	Fibula	IIA	CT, WR	NECO-95J	-	-	165	CDF	-	-	-	-
13	28/F	Humerus	IIA	CT, WR	NECO-95J	-	+	15	DOD	-	+	-	-
14	32	Pelvis	IIB	CT, HPR		-	-	55	AWD	+	+	-	-
15	28	Pelvis	IIB	CT, HPR	NECO-95J	-	-	10	DOD	+	+	-	+
16	63	Femur	IIA	WR	None	+	+	13	DOD	-	+	-	+

y: years, F: female, M: male, CT: chemotherapy, WR: wide resection, MR: marginal resection, mo: month(s), CDF: continuously disease-free, NED: no evidence of disease, DOD: dead of disease, AWD, alive with disease, IA: doxorubicin/ifosfamide, HPR: heavy particle radiation.

**Table 2 diagnostics-10-00528-t002:** Comparison of age, tumor size, and ALP between the PD-1-positive and -negative groups and between the PD-L1-positive and -negative group.

Clinical Parameter	PD-1		PD-L1		*p*-Value (PD-1/PD-L1)
	Positive	Negative	Positive	Negative	
Age (y.o)	36 ± 33.1	28 ± 16.4	32 ± 22.3	26 ± 13.9	0.53/0.56
Size (cm)	12 ± 15.5	15.5 ± 30.0	26.7 ± 35.3	58.5 ± 20.2	0.85/0.065
ALP (U/L)	273 ± 199.3	363.5 ± 294.5	266 ± 278.9	368 ± 274.7	0.63/0.53
SUV-max value	12.7 ± 6.56	10.5 ± 5.47	11.1 ± 5.51	11.3 ± 1.15	0.62/0.48

ALP: Alkaline phosphatase, y.o: years old, SUV: standardized uptake value.

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
