# Peer review of "Characterization of PD-1/PD-L1 Immune Checkpoint Expression in Osteosarcoma"

_diagnostics, 2020, doi:10.3390/diagnostics10080528_

Round 1
Reviewer 1 Report
The manuscript "Characterization of PD-1/PD-L1 Immune Checkpoint Expression in Osteosarcoma” by Hashimoto et al reports using IHC staining to identify CD4, CD8, PD-1 and PD-L1 expression in 16 osteosarcoma patients. They concluded that a poor outcome was associated with the PD-1-negative and PD-L1-positive status. This is very interesting topic, because the 5-year survival rate is still dismal. Therefore, to find new therapy to elevate prognosis is urgently needed. However, there are some questions that need to be addressed.
- In their conclusion, PD-L1 expression was associated with poor outcome. According to reference 18, tumor size is an important prognostic factor in osteosarcoma. In line 166, the authors mentioned “In this study, PD-L1-positive patients had significantly larger tumor sizes than PD-L1-negative patients.”. However, in Figure 2D, PD-L1-negative tumors are larger than the PD-L1-positive tumors (p=0.03). The data is not consistent to their conclusion. This is very important issue in this paper, because this is the only one significant result in this study.
- The IHC staining is not clear in figure 1. I suggested to show larger microscopic view. Furthermore, PD-1 is mainly expressed in T cells and pro-B cells. I suggested them to show same view which are stained by CD4, CD8, and PD-1.
- In this study, they discussed CD4, CD8, PD-1 and PD-L1 expression in osteosarcoma. However, previous studies, such as reference 9 and 10, were reported this issue in osteosarcoma, even in all sarcoma. Therefore, this study is less novelty.
Author Response
Reviewer#1
Thank you for taking the time to review this article. We have revised this paper in accordance with your helpful comments.
We believe this paper has benefited from your constructive comments, and the revised results and conclusions are now consistent and more transparent.
Please refer to our point-by-point responses.
Comments and Suggestions for Authors
The manuscript “Characterization of PD-1/PD-L1 Immune Checkpoint Expression in Osteosarcoma” by Hashimoto et al reports using IHC staining to identify CD4, CD8, PD-1 and PD-L1 expression in 16 osteosarcoma patients. They concluded that a poor outcome was associated with the PD-1-negative and PD-L1-positive status. This is very interesting topic, because the 5-year survival rate is still dismal. Therefore, to find new therapy to elevate prognosis is urgently needed. However, there are some questions that need to be addressed.
- In their conclusion, PD-L1 expression was associated with poor outcome. According to reference 18, tumor size is an important prognostic factor in osteosarcoma. In line 166, the authors mentioned “In this study, PD-L1-positive patients had significantly larger tumor sizes than PD-L1-negative patients.”. However, in Figure 2D, PD-L1-negative tumors are larger than the PD-L1-positive tumors (p=0.03). The data is not consistent to their conclusion. This is very important issue in this paper, because this is the only one significant result in this study.
Author’s response
Thank you for pointing out that.
When we double-checked our statistical methods, we noted that in the figure’s caption, we mistakenly indicated that a one-sided test was used. Instead, a two-tailed test was used, which showed a p = 0.065. Therefore, no statistically significant difference was found.
Nevertheless, even without a statistically significant difference, PD-L1-positive tumors tended to be larger.
Although there is no relationship between survival and the PD-1/PD-L1 immune checkpoint, the cases characterized by recurrence, metastasis, and death were PD-1-negative and PD-L1-positive. Therefore, we believe that the PD-1 negative and PD-L1 positive status is involved in the occurrence of cancer-related events.
Author’s action:
The relevant sentences in the results, discussion, and abstract were revised as follows.
In the results part:
Tumor sizes were 12 ± 15.5 cm in the PD-1-positive group (n = 3, Figure 2cTable 2) and 15.5 ± 30.0 cm in the PD-1-negative group (n = 10, Figure 2cTable 2), and the difference was not significant (p = 0.85). Tumor sizes were 26.7 ± 35.3 cm in the PD-L1-positive group (n = 10, Figure 2dTable 2) and 58.5 ± 20.2 cm in the PD-L1-negative group (n = 6, Figure 2dTable 2), and the difference was not significant (p = 0.030.065). (Line 84-88)
In the discussion part:
In the current study, no significant difference in tumor size was observed between PD-1-positive and -negative cases or between PD-L1-positive and -negative cases. (Line 191-194)
These findings indicate that a PD-1-negative and PD-L1-positive status may be associated with aworse outcome in osteosarcoma. These findings indicate that, in OS, a PD-1-negative and PD-L1-positive status may be associated with the occurrence of cancer-related events, such as recurrence, metastasis, and death. (Line 216-219)
We believe that the PD-1/PD-L1 immune checkpoint system, involving CD4 and CD8, plays an important role in the pathogenesis of osteosarcomaOS and a PD-1-negative with PD-L1-positive status may predict the occurrence of cancer-related eventsa poor outcome. (Line 221-224)
In the abstract:
Age, size, ALP, and SUV-max levels did not differ significantly between PD-1/PD-L1-positive and -negative cases. The 3-year survival rates did not differ significantly between PD-1-positive and -negative cases or between PD-L1-positive and -negative cases. AHowever, the occurrence of cancer-related events, including recurrence, metastasis, and death poor outcome, was associated with the PD-1-negative and PD-L1-positive status. The PD-1/PD-L1 checkpoint is likely involved in the immune microenvironment in osteosarcomaOS and may be a prognostic indicator involved in the occurrence of cancer-related events. (Line 18-24)
- The IHC staining is not clear in figure 1. I suggested to show larger microscopic view. Furthermore, PD-1 is mainly expressed in T cells and pro-B cells. I suggested them to show same view which are stained by CD4, CD8, and PD-1.
Author’s response
Thank you very much for pointing this out. As you commented, Figure 1 was a bit blurry and unclear. We believe that a larger and clearer micrograph should be displayed.
We also agree that results should be presented in a consistent way, similar to that of CD4, CD8, and PD-1 staining.
Author’s action:
We revised Figure 2 and added a caption.
- In this study, they discussed CD4, CD8, PD-1 and PD-L1 expression in osteosarcoma. However, previous studies, such as reference 9 and 10, were reported this issue in osteosarcoma, even in all sarcoma. Therefore, this study is less novelty.
Author’s response
Thank you for the suggestion.
As you pointed out, although several published papers suggested the involvement of the PD-1/PD-L1 immune checkpoint mechanism in osteosarcoma, to the best of our knowledge, we believe that this is the first study to compare the clinicopathological details.
We believe that the novelty of this study should be emphasized.
Author’s action:
We revised the following sentences in the abstract and discussion section.
In the abstract:
AHowever, the occurrence of cancer-related events, including recurrence, metastasis, and death poor outcome, was associated with the PD-1-negative and PD-L1-positive status. The PD-1/PD-L1 checkpoint is likely involved in the immune microenvironment in osteosarcomaOS and may be a prognostic indicator involved in the occurrence of cancer-related events. (Line 20-24)
In the discussion part:
These findings indicate that a PD-1-negative and PD-L1-positive status may be associated with aworse outcome in osteosarcoma. These findings indicate that, in OS, a PD-1-negative and PD-L1-positive status may be associated with the occurrence of cancer-related events, such as recurrence, metastasis, and death. (Line 216-219)
In conclusion, we surveyed and characterized the expression of CD4, CD8, PD-1, and PD-L1 in osteosarcomaOS cases treated in our hospital in detail. We believe that the PD-1/PD-L1 immune checkpoint system, involving CD4 and CD8, plays an important role in the pathogenesis of osteosarcomaOS and a PD-1-negative with PD-L1-positive status may predict the occurrence of cancer-related eventsa poor outcome. (Line 221-224)

Reviewer 2 Report
The manuscript by Hashimoto and colleagues evaluate the relantionship between clinical factors and the expression of PD-1 and PD-L1 in OS.
Points to be addressed:
-Lines 26/27. The sentence " They originate from bone and, rarely, from soft tissue." is not clear. Please modiy
-The abbreviated form OS should be used throughout the manuscript.
-Line 31 please change sarcoma in OS
-Lines 66-108 Results from statistical analysis of clinical and histological data should be reported as table.
-Figure 2 should be removed.
Author Response
Reviewer#2
Thank you for taking the time to review this article. We have revised this article in accordance with the helpful comments that accompany your deep insights. Please refer to our point-by-point responses.
Comments and Suggestions for Authors
The manuscript by Hashimoto and colleagues evaluate the relationship between clinical factors and the expression of PD-1 and PD-L1 in OS.
Points to be addressed:
-Lines 26/27. The sentence “They originate from bone and, rarely, from soft tissue.” is not clear. Please modify
Author’s response
Thank you for pointing this out. The sentence was unclear and should be revised for clarity.
Author’s action:
We revised the sentence as follows.
They are malignant bone tumors that primarily originate from the bone marrow, and rarely from soft tissue. (Line 28-30)
-The abbreviated form OS should be used throughout the manuscript.
Author’s response and action
We agree and made sure to use the abbreviation OS throughout the revised manuscript.
-Line 31 please change sarcoma in OS
Author’s response and action
We agree and changed “sarcoma” to OS.
-Lines 66-108 Results from statistical analysis of clinical and histological data should be reported as table.
Author’s response and action
Thank you for this comment. We agree that these results should be presented in a Table.
Results from the statistical analysis of clinical and histological data are now shown in Table 2.
Accordingly, Figure 2 has been revised.

Reviewer 3 Report
The manuscript is clearly written and easily readable. The major limitation is the small number of patients analyzed. In my opinion the limited number of patients likely weakens the statistic results. In particular, PD-1 positive patients are only 3.
The conclusions do not are compliant with the results. in my opinion the manuscript is not acceptable in the present form and major revision is nedeed.
Then the figure legend of Figure 2 must be shortened, the authors repeated all the results, but it is redundant.
Major points:
- In the discussion lines 150-151, pag 7 the authors wrote: In the current study, there was a strong correlation between CD4 positivity or CD8 positivity and PD-L1. Actually, in this study there is not a correlation analysis between CD4+ or CD8+ and PD-L1
- In the discussion lines 165-166, pag 7 the authors wrote: In this study, PD-L1-positive patients had significantly larger tumor sizes than PD-L1-negative patients. PD-L1-positive osteosarcoma cases may have a poorer prognosis. The Fig.3 shows exactly the opposite, so the conclusion about PD-L1 expression as prognostic factor does not seem consistent with the results.
- it is needed to check references: 24 and 26 is the same reference.
Author Response
Reviewer#3
Thank you for taking the time to review this article. As you pointed out, in this study, we analyzed a relatively small cohort. Besides, there were a few discrepancies between the results and the conclusions. Furthermore, the legend of Figure 2 was redundant, and we agree that it should be revised.
We have followed your comments and revised the text accordingly. We believe that the paper has benefited from your valuable insight. Please refer to our point-by-point responses.
Comments and Suggestions for Authors
The manuscript is clearly written and easily readable. The major limitation is the small number of patients analyzed. In my opinion the limited number of patients likely weakens the statistic results. In particular, PD-1 positive patients are only 3.
The conclusions do not are compliant with the results. in my opinion the manuscript is not acceptable in the present form and major revision is nedeed.
Then the figure legend of Figure 2 must be shortened, the authors repeated all the results, but it is redundant.
Major points:
- In the discussion lines 150-151, pag 7 the authors wrote: In the current study, there was a strong correlation between CD4 positivity or CD8 positivity and PD-L1. Actually, in this study there is not a correlation analysis between CD4+ or CD8+ and PD-L1
Author’s response:
Thank you for your comment. However, we have already investigated the correlation between CD4+ or CD8+ and PD-L1, as quoted below:
“The CD4-positive and PD-L1-positive rate was 9/16 (56.2%). The CD8-positive and PD-L1-positive rate was 8/16 (50%). The PD-1 and PD-L1 concordance rate was 3/16 (18.7%).” (Line 61-63)
- In the discussion lines 165-166, page 7 the authors wrote: In this study, PD-L1-positive patients had significantly larger tumor sizes than PD-L1-negative patients. PD-L1-positive osteosarcoma cases may have a poorer prognosis. The Fig.3 shows exactly the opposite, so the conclusion about PD-L1 expression as prognostic factor does not seem consistent with the results.
Author’s response:
Thank you for pointing out the discrepancy between results and conclusion.
As you pointed out, the tumor size is larger in PD-L1-positive tumors, and survival may be worse in the PD-L1-positive group.
We reviewed the statistical methods used and found that the p-values returned by the one-sided test were mistakenly reported.
The results of the two-tailed test were found not to be significantly different (p = 0.065). Nevertheless, the tumor size may be larger in the PD-L1-positive group.
Therefore, we believe that we should revise our conclusions.
Although there is no relationship between survival and the PD-1/PD-L1 immune checkpoint, the cases characterized by recurrence, metastasis, and death were PD-1-negative and PD-L1-positive. Therefore, we believe that the PD-1 negative and PD-L1 positive status is involved in the occurrence of cancer-related events.
Author’s action:
We revised relevant sentences in the results and discussion sections, as follows.
In the results part:
Tumor sizes were 12 ± 15.5 cm in the PD-1-positive group (n = 3, Figure 2cTable 2) and 15.5 ± 30.0 cm in the PD-1-negative group (n = 10, Figure 2cTable 2), and the difference was not significant (p = 0.85). Tumor sizes were 26.7 ± 35.3 cm in the PD-L1-positive group (n = 10, Figure 2dTable 2) and 58.5 ± 20.2 cm in the PD-L1-negative group (n = 6, Figure 2dTable 2), and the difference was not significant (p = 0.030.065). (Line 84-88)
In the discussion part:
In this study, PD-L1-positive patients had significantly larger tumor sizes than PD-L1-negative patients. PD-L1-positive osteosarcoma cases may have a poorer prognosis. In the current study, no significant difference in tumor size was observed between PD-1-positive and -negative cases or between PD-L1-positive and -negative cases. (Line 191-194)
These findings indicate that a PD-1-negative and PD-L1-positive status may be associated with aworse outcome in osteosarcoma. These findings indicate that, in OS, a PD-1-negative and PD-L1-positive status may be associated with the occurrence of cancer-related events, such as recurrence, metastasis, and death. (Line 216-219)
We believe that the PD-1/PD-L1 immune checkpoint system, involving CD4 and CD8, plays an important role in the pathogenesis of osteosarcomaOS and a PD-1-negative with PD-L1-positive status may predict the occurrence of cancer-related eventsa poor outcome. (Line 221-224)
- it is needed to check references: 24 and 26 is the same reference.
Author’s response and action:
Thank you for your valuable comment. We apologize for the missing citation. We revised the reference list and amended the reference numbers.

Round 2
Reviewer 1 Report
Thanks for kindly replying.
Only one typo need to be fixed.
Line 81, Figure 2a -> Table 2.
Author Response
Thank you for taking the time to review this article again.
We have corrected that in accordance with your useful comment.
We would appreciate it if you would review that again.
Reviewer 1:Thanks for kindly replying.
Only one typo need to be fixed.
Line 81, Figure 2a -> Table 2.
Author’s response and action:
Thank you for your pointing out our typo. Also, we apologize for forgetting to correct this. The “Figure 2a” is correctly Table 2. So, we revised “Figure 2a” to “Table 2”.
Reviewer 3 Report
"As you pointed out, the tumor size is larger in PD-L1-positive tumors, and survival may be worse in the PD-L1-positive group.
We reviewed the statistical methods used and found that the p-values returned by the one-sided test were mistakenly reported.
The results of the two-tailed test were found not to be significantly different (p = 0.065). Nevertheless, the tumor size may be larger in the PD-L1-positive group."
How can the authors declare that the tumor size is larger in PD-L1-positive tumors, if in table 2 it is reported tumor size of PD-L1 positive 26.7 ±
35.3 vs tumor size of PD-L1 negative 58.5±20.2 ?
26,7cm < 58,5cm....thus the tumor size is larger in PD-L1 negative tumors, is it right?
Author Response
Thank you for taking the time to review this article again.
We are deeply grateful for the detailed review.
We have made reply comment in accordance with your useful comment.
We would appreciate it if you would review that again.
"As you pointed out, the tumor size is larger in PD-L1-positive tumors, and survival may be worse in the PD-L1-positive group.
We reviewed the statistical methods used and found that the p-values returned by the one-sided test were mistakenly reported.
The results of the two-tailed test were found not to be significantly different (p = 0.065). Nevertheless, the tumor size may be larger in the PD-L1-positive group."
How can the authors declare that the tumor size is larger in PD-L1-positive tumors, if in table 2 it is reported tumor size of PD-L1 positive 26.7 ±
35.3 vs tumor size of PD-L1 negative 58.5±20.2 ?
26,7cm < 58,5cm....thus the tumor size is larger in PD-L1 negative tumors, is it right?
Author’s response and action:
Thank you for your pointing out the errors in “PD-L1 positive group” and “PD-L1 negative group”.
We agree with your comment and apologize for mistaking “PD-L1 negative group” to “PD-L1 positive group”.
Tumor size is larger in the PD-L1 negative group, as shown in Table 2. And there was no significant difference in the two-tailed test.
This is simply an error on our description.
Therefore, the claim we want to make remains unchanged, and we have not revised the text in this time. The revision is as previously made.
We sincerely apologize for some confusion in your peer review.